# Dental Anxiety and Higher Sensory Processing Sensitivity in a Sample of German Soldiers with Inflammatory Periodontal Disease

**DOI:** 10.3390/ijerph18041584

**Published:** 2021-02-08

**Authors:** Thomas Eger, Felix Wörner, Ursula Simon, Sandra Konrad, Anne Wolowski

**Affiliations:** 1Department of XXIII Dentistry-Periodontology, Bundeswehr Central Hospital Koblenz, Ruebenacherstrasse 170, 56072 Koblenz, Germany; FelixWoerner@bundeswehr.org; 2Department of VI Center for Mental Health and Psychiatry, Bundeswehr Central Hospital Koblenz, Ruebenacherstrasse 170, 56072 Koblenz, Germany; UrsulaSimon@bundeswehr.org; 3Department of Personality Psychology and Psychological Diagnostics, Faculty of Humanities and Social Sciences, Helmut-Schmidt-University/University of the Bundeswehr Hamburg, Gebäude H4, Holstenhofweg 85, 22043 Hamburg, Germany; Konrads@hsu-hh.de; 4Department of Prosthodontics and Biomaterials, Albert-Schweitzer-Campus 1, University Hospital and Faculty of Medicine Muenster, 48149 Muenster, Germany; wolowsk@uni-muenster.de

**Keywords:** periodontitis, dental anxiety, sensory processing sensitivity, military dental fitness

## Abstract

(1) Background: Dental anxiety with disease value usually leads to avoidance of dental treatment. For the initial diagnosis of the level of anxiety, questionnaires such as the Hierarchical Anxiety Questionnaire (HAQ) are suitable. The construct of sensory processing sensitivity (SPS) describes a general trait in which people with a higher degree of SPS perceive information more strongly and process it more thoroughly. (2) Methods: This cross-sectional study evaluated the relationship between dental anxiety and higher levels of SPS in 116 soldiers referred with different stages of periodontitis for mandatory dental fitness before military deployment. (3) Results: The proportion of patients with periodontitis in stage III + IV was 39% and in stage I + II was 27%. The mean cumulative values of the questionnaires were 20.9 ± 10.6 for HAQ and 27.7 ± 16.0 for SPS. Eleven moderately anxious patients had a SPS value of 37.4 ± 13.5 and 10 highly anxious patients had a value of 36.3 ± 14.1. Patients diagnosed with stage III + IV periodontitis showed significantly higher values on the SPS subscale Low Sensory Threshold (LST), which describes overstimulation by external sensory stimuli, compared to patients with stage I + II periodontitis. Dental anxiety showed moderately significant correlations with the SPS subscale Ease of Excitation (EOE), which measures emotional reactivity to physiological stimuli. (4) Conclusions: Due to the frequency of dental anxiety and higher sensitivity in patients with severe periodontitis, it is useful to record said frequency.

## 1. Introduction

Periodontal disease is a pandemically non-communicable disorder [1,2] with serious socioeconomic consequences and a high burden on quality of life [3,4,5]. It is a chronic multifactorial inflammatory disease associated with dysbiotic dental plaque biofilms [6]. Periodontitis is characterized by the progressive destruction of the tooth-supporting apparatus [7]. If untreated, it may lead to tooth loss, although it is preventable and treatable in the majority of cases [7]. Furthermore, periodontitis is a risk factor for the development of osteonecrosis of the jaws in patients taking antiresorptive drugs [8]. During periodontal treatment, patients must know and comprehend what a periodontal disease is and why therapeutic adherence is the key to initial and long-term treatment success [9,10].

The development of dental anxiety is considered a multifactorial event [11,12]. In adults, dental anxiety does not usually occur spontaneously and unexpectedly. Those affected usually report a beginning in childhood or as adolescents and suffer with every upcoming visit to the dentist from the appearance of stressful anxiety feelings that are difficult to bear [12,13]. The influence of the following factors is considered empirically well documented: traumatic experiences, family influences, and individual characteristics (e.g., vulnerability). Beaton et al. [14] call the first two (according to Weiner and Sheehan [15]) exogenous factors because they affect the individual from the outside, while individual characteristics constitute the endogenous factors. Dental anxiety is classified in the clinical classification systems as a specific phobia (ICD-10 F40.2, fear of specific situations, here: medical contexts). These phobias are characterized by intense fear during treatment or its avoidance, combined with a clear sense of suffering and the occurrence of at least two of the known fear symptoms. “Specific” describes the fear of dental treatment with disease value, since these symptoms are limited to the dental treatment context [16,17]. The further criterion “insight that the fear is exaggerated or unreasonable” is used to distinguish phobic expressions from an intense anxiety experience. However, those affected often do not recognize that their anxiety is “unreasonable”—they cite the actual risks of treatment (pain, unpleasant sensations, medical complications) as reasons for their anxiety. The prevalence of dental anxiety with disease value is about 5–10% of the population, and dental treatment is usually avoided by these people [18]. Dental anxiety in hospitalized soldiers with post-traumatic stress disorder (PTSD) after military deployments is highly prevalent [19]. This fear often stands in the way of successful dental care for patients and is therefore a clinically relevant problem. For the initial diagnosis of the level of anxiety, questionnaires such as the Hierarchical Anxiety Questionnaire (HAQ) [20] are suitable. The dentist can suspect a diagnosis of dental anxiety with disease value if the patient is very anxious (HAQ cumulative score > 38) and at the same time avoids being treated for more than 2 years [21]. In a study by Lenk et al., using the HAQ, it was shown that in 212 patients with a psychosomatic illness, every third patient showed a pathologically high anxiety of dental treatment [22].

The construct of sensory processing sensitivity (SPS) describes a general trait in which people with a higher degree of SPS perceive information more strongly and process it more thoroughly [23]. It is assumed that this sensitive perception easily leads to overstimulation or hyperexcitability, especially when several pieces of information have to be processed simultaneously [24]. Furthermore, it is assumed that individuals with higher levels of SPS show stronger emotional responses and are more empathetic. Four indicators associated with higher sensory processing sensitivity are presented: (1) behaviour inhibition/avoidance, (2) more thorough information processing, (3) sensitivity to stimuli, and (4) emotional/physiological reactivity [24]. This characteristic is associated with a higher vulnerability to mental disorders. The relationship between SPS, mental disorders (e.g., anxiety, depression), and mental health has been well studied [25,26,27,28,29,30]. Studies on the association between SPS and physical diseases have also been conducted (e.g., type 1 diabetes) [31]. No SPS mean values for soldiers or other professional groups are available.

The aim of the present study in soldiers with inflammatory periodontal disease is to evaluate a possible relationship between dental anxiety and higher levels of SPS in patients with varying degrees of periodontal disease. Furthermore, it should be examined whether SPS or the subfactors (Ease of Excitation (EOE), Aesthetic Sensitivity (AES), Low Sensory Threshold (LST)) are an indicator for a periodontal disease diagnosis.

## 2. Material and Methods

This cross-sectional study was conducted at the Bundeswehr Medical Service Academy (Munich, Germany) and registered in the military clinical trial register (23K1-S-80 1921). All soldiers referred for periodontal reasons over the span of 8 months and who received treatment before expected military deployment within 4 months were recruited.

All patients were examined by a periodontist (TE) between October 2019 and May 2020 at the Bundeswehr Central Hospital Koblenz, Germany. Exclusion criteria were a history of PTSD or pregnancy.

The periodontal diagnosis was made according to the classification scheme defined in the 2017 World Workshop on the Classification of Periodontal and Peri-Implant Diseases and Conditions [32]. Examining a new patient consisted of four sequential steps [7]: Identifying a patient suspected of having periodontitis, confirming the diagnosis of periodontitis, staging the periodontitis case, and grading the periodontitis case. For full-mouth periodontal probing, clinical attachment loss, and bleeding on probing measurements at six sites per tooth, a pressure calibrated manual periodontal probe was used (Aesculap DB764R, Tuttlingen, Germany).

Self-administered questionnaires for smoking habits, HAQ, which also addresses different treatment situations with 11 items, and a scale for testing higher sensory processing sensitivity (HSPS) with 26 items were used.

The HAQ is based on the Dental Anxiety Scale according to Corah [33] and also contains six different treatment situations, which are taken from the anxiety hierarchy of a study by Gale [34] and represent the most anxiety-inducing situations in patient treatment. The HAQ consists of 11 questions in which five different levels of anxiety can be selected (from “not anxious at all” to “sick with anxiety”); thus, a score from 11 to 55 is possible. Patients can be divided into three groups: low anxiety (up to 30 points), medium anxiety (from 31 to 38 points), and high anxiety (more than 38 points). In addition, the HAQ can be used to derive the suspected diagnosis of dental anxiety with disease value if a score of more than 38 points is reached while at the same time dental treatment is avoided for more than two years [35].

SPS is measured with HSPS, which has also been validated for German-speaking countries [23,25]. The HSPS is a 26-item scale with a 5-level response format (from “0” representing “does not apply at all” to “4” representing “fully applies”). SPS is a multidimensional construct consisting of the factors EOE, AES, and LST. EOE measures emotional reactivity to physiological stimuli, AES stands for deeper processing in the sense of stronger reflection, and LST describes overstimulation by external sensory stimuli. These three factors are relatively highly correlated with each other, which suggests an overriding characteristic, namely SPS. The scale has good to very good reliabilities and has been scientifically validated [25].

The statistical analysis was performed using the statistical software SPSS 24 (IBM Corp., Armonk, NY, USA) for descriptive presentation, bivariate correlations, and regression data analysis. Descriptive data are presented as the means, standard deviation, or number (percentage). Regression analysis were used to test how well the variable SPS is able to predict the diagnosis of periodontitis. Possible confounding variables, such as dental anxiety, stages of periodontitis, location, number of teeth, smoking as a risk factor, and gender, were considered. Normal distribution was tested with the Kolmogorov–Smirnov test. This was rejected for most variables (except EOE, AES, LST, HSPS_GS). Therefore, non-parametric procedures were used. Spearman correlation was calculated for the correlation analyses. The EOE, AES, and LST subscales of the HSPS are highly correlated with each other, so partial correlation was used here to avoid spurious correlations. To determine predictors of dental anxiety (HAQ), a stepwise regression was first calculated using the HSPS subscales (EOE, AES, LST) as predictor variables. The next step was to test which predictors best predicted the diagnosis of periodontitis. The following variables were included in the model as predictors: HAQ cumulative score (HAQ_GS), number of teeth (not including 8th teeth), number of cigarettes, duration of smoking, EOE, AES, LST, and HSPS cumulative score (HSPS_GS). Stepwise regression was also chosen for this purpose. In this procedure, the most significant variable is systematically added automatically, and the least significant variable is removed. Incomplete datasets were excluded. This procedure was chosen because no empirical evidence was yet available on which to base the analysis. The significance level was set at *p* < 0.05.

In full accordance with ethical principles, the guidelines of the Helsinki Declaration were followed and the Regional Ethics Review of the State Chamber of Physicians of Rhineland-Palatinate in Germany (2019-14303) approved the study (26 July 2019).

Written informed consent was obtained from all subjects involved in the study after written information was provided to all referred patients. Subjects were informed that they could refrain from the study at any time without any consequence.

## 3. Results

One hundred and sixteen outpatients (41 women, 75 men, mean age 38.5 ± 12.6 years, 32% smokers, mean number of teeth 26.3 ± 2.7) were referred for periodontal reasons and mandatory dental check-up to the Department of Periodontology of the Bundeswehr Central Hospital Koblenz, Germany. The proportion of patients with stage III + IV periodontitis was 39%, with stage I + II was 27%, and with gingivitis/recessions was 34% (Table 1). A rapid progression rate for periodontitis (Grade C) was determined in 22 patients.

The cumulative scores of the questionnaires were 20.9 ± 10.6 for HAQ and 27.7 ± 16 for HSPS. Eleven patients had a cumulative HAQ value of 31–38 (moderately anxious) with a cumulative HSPS value of 37.4 ± 13.5, and 10 patients had a cumulative HAQ value >38 (highly anxious) with a cumulative HSPS value of 36.3 ± 14.1 (Table 2 and Table 3). The proportion of moderately and highly anxious patients in patients with stage III + IV periodontitis was 17%.

The HAQ showed moderately significant correlations with the SPS subscales EOE and LST and with the HSPS_GS. After partialing out the other subscales due to the described high intercorrelations, the correlation was only maintained with the EOE scale. In particular, the subscale LST and the diagnosis of periodontitis were associated with the severity of the periodontitis. The number of teeth showed moderately negative correlations with the overall score of EOE, LST, and HSPS-GS, the diagnosis of periodontitis, and the severity of the latter. The number of cigarettes and the smoking duration was weakly to moderately associated with the diagnosis of periodontitis and its severity. In addition, the duration of smoking was weakly associated with the LST subscale. All the results of the correlations can be seen in Table 4. A regression analysis to predict the HAQ showed that only the EOE subscale predicts dental anxiety (Table 5). The variance explanation here was 22.1%. The stepwise regression analysis showed that the number of teeth, the duration of smoking, and the LST are the best predictors of periodontal disease and explain 27.9% of the variance (Table 5). These results show that the subscale LST seems to be of special importance. Therefore, an additional univariate variance analysis was calculated to investigate the influence of the different periodontitis stages on the low sensory threshold. The factor was the periodontitis groups (1 = control group with gingivitis, recessions, 2 = periodontitis stage I + II for mild periodontitis, 3 = periodontitis stage III + IV for severe periodontitis), and low sensory threshold was the dependent variable. The Levene test was not significant. The results show a significant group effect *F*(2, 113) = 3.36, *p* = 0.038, *ηp*^2^ = 0.056. Post-hoc analyses using Tukey test show a significant group difference (*p* = 0.03) between patients with mild periodontitis (*MW* = 31.83, *SD* = 7.37) and patients with severe periodontitis (*MW* = 35.58, *SD* = 7.72). The effect size is *d* = 0.50 and corresponds to a medium effect according to Cohen.

## 4. Discussion

Patients diagnosed with stage III + IV periodontitis showed significantly higher values on the LST subscale, which describes overstimulation by external sensory stimuli compared to patients diagnosed with stage I + II periodontitis. Dental anxiety measured using the HAQ scale showed moderately significant correlations with the SPS subscale EOE, which measures emotional reactivity to physiological stimuli. The results showed that the diagnosis of periodontitis was associated with a low sensory threshold (LST) but also with the HSPS-G overall score. Dental anxiety was in turn associated with high reactivity to internal or external stimulation (EOE) and LST. In this case, the LST subscale showed a pseudo-correlation, as after the other subscales were separated, the correlation was not maintained. EOE indicated dental anxiety and explained 22.1% of the variance of dental anxiety. All other subscales did not contribute to the explanation of the variance of dental anxiety.

These results were consistent with results from other studies. Correlations have been reported between higher SPS levels and higher levels of perceived stress and common disease symptoms, but perceived stress levels do not moderate health. The relationship between SPS and physical symptoms could only be attributed to the subscales EOE and LST [26]. These subscales have also shown significant correlations between the three SPS factors and anxiety, and between EOE and depression, as well as a high correlation between neuroticism and physical symptoms, anxiety, depression, low mental health and EOE [26]. In a study by Konrad and Herzberg, all nine scales (aggressiveness, anxiety, depression, paranoid thinking, phobic anxiety, psychoticism, somatization, insecurity in social contact, obsessive-compulsivity) of the Brief Symptom Checklist were weakly to moderately associated with EOE after the other factors had been separated [25]. EOE has also been associated with the 5-HTTLPR polymorphism (5-HTT gene linked polymorphic region) [36]. This serotonin transporter polymorphism in turn has been associated with higher levels in neuroticism, a personality trait that also plays a role in SPS and has been associated with anxiety [23,26].

Another finding of the present study was that the number of teeth, the duration of smoking, and the LST were the best predictors of periodontal disease and explained 27.9% of the variance. A group comparison showed that patients diagnosed with stage III + IV periodontitis (values 6–9) reported significantly higher values on the LST subscale compared to patients diagnosed with stage I + II periodontitis (values 2–5). This effect was moderately pronounced. This result is consistent with studies on the association between SPS and physical disease. In 2018, Goldberg et al. compared a type 1 diabetes group with a healthy control group with regard to their SPS values. The type 1 diabetes group reported significantly higher SPS values compared to the healthy control group. They cited the sympathetic nervous system, which is highly activated in autoimmune diseases, as the reason for the reported results [31].

Periodontitis is highly prevalent worldwide. Personalised periodontal care requires clinicians to understand the impact of periodontitis on patients’ daily lives. Periodontitis has a profound detrimental effect on patients’ psychosocial well-being. Periodontal treatment not only improves the physical symptoms, but can bring powerful improvements in attitude, self-esteem, mood, and social well-being. These insights may facilitate the delivery of periodontal care which responds to patients’ needs, thus improving their satisfaction, motivation, and adherence. Further, the psychosocial benefits of periodontal treatment may be useful to promote periodontal therapy and reassure those at the outset of treatment [37].

Many aspects of a dental treatment situation can cause anxiety, tension, and discomfort. However, these elements do not lead to pathological dental anxiety in every person. Most people learn to deal with regular visits to the dentist without restrictions despite these unpleasant conditions. However, there are certain factors that make this adjustment unsuccessful and instead lead to the development of excessive and harmful anxiety. In a demographic survey, 300 residents of a German city were questioned to determine the prevalence of dental anxiety. The HAQ was used to measure the amount of dental anxiety. The average level of anxiety was 28.8 (*SD* = 10.1) according to the HAQ. Young people were more anxious than older people (*p* = 0.007), and women were more anxious than men (*p* = 0.004). Of all participants, 11% [95% CI: (7.5%; 14.5%)] suffered from dental phobia [21]. In our study, 10 patients (9%) with periodontitis were found to be highly anxious with dental avoidance behaviour. These patients were presented with a lack of military dental fitness and possible negative consequences for their further professional career. Similar results of 10.9% were found for 374 German soldiers at the age of 19–29 years. On their compulsory dental check-up, community periodontal treatment needs values of anxious and less anxious patients showed no differences [38]. In another study, 12% of 176 patients referred for periodontal therapy to a Norwegian specialist private practice reported extreme anticipatory anxiety [39]. Females recorded significantly higher anxiety scores than males [39]. For periodontal surgery and implant treatments, pain perception was affected by the level of presurgical anxiety [40]. In a study of almost 6000 people representative for Finland, Pohjola et al. found that anxiety and depressive disorders were more common in the group of highly dental anxious (determined to be “very anxious” according to a self-assessment with a question where participants could select not/slightly/very for their perceived anxiety level) people [41]. They also found increased dental fear in patients with depression and anxiety disorders, with the highest prevalence in the group of combined depression and anxiety disorders [41]. Connections between anxiety of dental treatment, depression, and anxiety were also confirmed by Bernson et al. [42]. Fear of dental treatment is an underrecognized symptom in people with impaired mental health [22]. Lenk et al. found increased anxiety of dental treatment (measured with the HAQ), compared to healthy control persons, in 30.5% of 212 patients in a psychosomatic clinic. In patients with post-traumatic stress disorder (from sexual abuse), increased anxiety of dental treatment was found to be even more frequent, with it being observed in 42.0% of patients [22]. Increased dental anxiety (measured by the Dental Fear Survey) was also found in patients with attention deficit syndrome compared to healthy individuals [43].

### Limitations of the Study

This study included only soldiers as patients. Soldiers undergo mandatory dental examinations by the military to determine their dental fitness. Dental treatment, on the other hand, is mandatory only for deployment. Soldiers are at higher risk for the development of PTSD after military deployment all over the world [19]. PTSD prevention efforts are therefore still needed. Although the sample size was small, all the soldiers referred for 8 months were recruited.

The mean value of dental anxiety among soldiers in our sample group might be lower than that in former studies on younger civilians and soldiers in Germany [21,38]. This highly specialized group could hardly be compared to the general population, especially when psychological variables were compared with clinical parameters. Military dentists’ empathy towards their patients’ stress reaction in regard to periodontal treatment needs for military deployments have not been analysed yet.

Soldiers in this study showed lower SPS averages compared to the normal population. Konrad and Herzberg reported in 2019 a mean value of 74.21 (*SD* = 16.85) for the HSPS overall score in their validation study [25]. The lower mean values could have been related to a Western world view, where it is negatively connotated for soldiers to stand by their sensitivity, or they actually have lower values and for this reason are more likely to choose the profession.

It has not yet been clarified whether burnout or other mental disorders after military deployments could be prevented by an early determination of the SPS and dental anxiety and, if necessary, resilience training or early intervention of the latter. Measurement of dental fear before deployment of soldiers might be integrated in pre-deployment resilience training.

## 5. Conclusions

The HAQ showed medium significant correlations with the SPS-subscale EOE, which measures emotional reactivity to physiological stimuli. Number of teeth, duration of smoking, and LST, which describes overstimulation by external sensory stimuli, were the best predictors of periodontal disease. Patients diagnosed with stage III + IV periodontitis showed significantly higher values on the LST subscale compared to patients diagnosed with stage I + II periodontitis.

Due to the frequency of dental anxiety and higher sensitivity in soldier patients with severe periodontal disease, it was useful to record the findings. To date, no data is available from prospective randomized studies on the changeability of both factors (EOE and LST) through cognitive behavioural therapy in periodontal therapy or during military deployment activities.

## Figures and Tables

**Table 1 ijerph-18-01584-t001:** Distribution of periodontitis stages including the risk factor of smoking in the patient population.

	*n*	Smokers	Women	Men
Recessions	7	29%	5	2
Gingivitis	33	18%	18	15
Periodontitis stage I localized	2	0	2	0
I generalized	7	29%	3	4
II localized	8	25%	3	5
II generalized	14	21%	3	11
III localized	25	60%	5	20
III generalized	11	36%	1	10
IV localized	9	33%	1	8

**Table 2 ijerph-18-01584-t002:** Cumulative scores of Hierarchical Anxiety Questionnaire (HAQ) and higher sensory processing sensitivity (HSPS) in the respective stages of periodontitis.

	*n*	HAQ		HSPS	
		Cumulative Score	*SD*	Cumulative Score	*SD*
Gingivitis/Recessions	40	15.1	11	33.6	16.8
Periodontitis stage					
I	9	17.5	6.5	24.4	16.2
II	22	24.9	11.0	27.2	16.7
III	36	21.6	11.4	27.1	15.8
IV	9	24.6	12.4	33.1	14.9

**Table 3 ijerph-18-01584-t003:** Cumulative scores of HSPS in the respective stages of dental fear (HAQ).

	HAQ		HSPS	
	Cumulative Score	*SD*	Cumulative Score	*SD*
HAQ average	20.9	10.6	27.7	16.0
HAQ cumulative score				
<30	*n*:95		25.7	15.1
31–38 (moderate anxiety)	*n*:11		37.4	13.5
>38 (high anxiety)	*n*:10		36.3	14.1

**Table 4 ijerph-18-01584-t004:** Bivariate correlations of sensitivity (SPS), HAQ, diagnosis of periodontitis, and risk factors (*n* = 116).

	1		2		3		4		5		6		7		8		9	
1. EOE *t*-value	1		0.38	**	0.75	**	0.91	**										
2. AES *t*-value	0.38	**	1		0.37	**	0.60	**										
3. LST *t*-value	0.75	**	0.37	**	1		0.88	**										
4. HSPS_GS *t*-value	0.91	**	0.60	**	0.88	**	1											
5. Diagnosis	0.19	*	0.16		0.35	**	0.27	**	1									
					0.32	**												
6. Grading	0.12		0.10		0.24	*	0.19	*	0.77	**	1							
7. Number of teeth w/o 8th	−0.30	**	−0.10		−0.36	**	−0.34	**	−0.46	**	−0.41	**	1					
8. Number of cigarettes	0.17		−0.06		0.11		0.08		0.23	*	0.23	*	−0.09		1			
9. Duration of smoking	0.09		0.04		0.21	*	0.13		0.35	**	0.38	**	−0.25	**	0.60	**	1	
10. HAQ_GS	0.48	**	0.08		0.44	**	0.44	**	0.22	*	0.10		−0.19	*	0.10		0.07	1
	0.27	**	−0.15		0.16													

Notes: 1 = Ease of Excitation (*t* value), 2 = Aesthetic Sensitivity (*t* value), 3 = Low Sensory Threshold (*t* value), 4 = Highly Sensitive Person Scale Global Score (*t* value), 5 = Diagnosis, 6 = Grading, 7 = Number of teeth w/o 8th, 8 = Number of cigarettes, 9 = Duration of smoking, 10 = Hierarchical Anxiety Questionnaire Global Score. ** Correlation is significant at 0.01 level (two-sided). * Correlation is significant at 0.05 level (two-sided). EOE: Ease of Excitation; AES: Aesthetic Sensitivity; LST: Low Sensory Threshold.

**Table 5 ijerph-18-01584-t005:** Regression analysis for predicting dental anxiety and periodontal diagnosis (*n*= 116).

Dependent Variable		Non-Standardized	Standarized		adj. *R*^2^
		*β*	*β*	*t*	
**HAQ**	**Model 1**				
	Ease of Excitation	0.61	0.48 ***	5.80	0.221
**Periodontal Diagnosis**	**Model 1**				
	Number of teeth w/o 8th	−0.45	−0.46 ***	−5.54	0.205
	**Model 2**				
	Number of teeth w/o 8th	−0.39	−0.40 ***	−4.78	0.257
	Duration of smoking	0.36	0.25 **	2.99	
	**Model 3**				
	Number of teeth w/o 8th	−0.33	−0.34 ***	−3.90	0.279
	Duration of smoking	0.33	0.23 **	2.74	
	LST	0.06	0.18 *	2.12	

Note: *** *p* ≤ 0.001. ** *p* ≤ 0.01. * *p* ≤ 0.05.

## Data Availability

The data presented in this study are available on request from the corresponding author.

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
