# Peer review of "Dental Anxiety and Higher Sensory Processing Sensitivity in a Sample of German Soldiers with Inflammatory Periodontal Disease"

_ijerph, 2021, doi:10.3390/ijerph18041584_

Round 1
Reviewer 1 Report
Dear author(s),
The study is well conducted and presented accordingly, I hope you consider the following points:
- In Abstract, please reformulate it in a structured form (Background, Methods, Results, and Conclusions) in order to make it easier for readers to get an idea of what you have done.
- In Abstract, please explain what LST and EOE refer to.
- In Introduction, the order of the reference needs to be fixed.
- In Introduction, (Line 52), please specify which version of ICD you had used. Is it ICD-11 or 10 or what exactly?
- In Methods, (Line 127), please explain how the questionnaire was conducted. Was it self-administered questionnaire? or is it filled by the investigators? In case it was filled by the investigators, please explain how you calibrated your investigators.
- In Results, (Line 140), please explain what does Group C refer to?
- In Discussion, please add to your study's limitations the issue of your limited sample size and how it may have led to weak correlation between some variables in your study. for example, the weak correlation between smoking frequency/duration and the periodontitis presence/severity.
- In Discussion, there's no need to re-explain the abbreviations of LST and EOE, because they should have been described earlier in the manuscript.
Regards,
Author Response
Response to Reviewer 1 Comments.
Point 1: In Abstract, please reformulate it in a structured form (Background, Methods, Results, and Conclusions) in order to make it easier for readers to get an idea of what you have done.
Response 1: We structured the abstract in 4 parts as recommended.
Point 2: In Abstract, please explain what LST and EOE refer to.
Response 2: Both abreviations were explained in the abstract as recommended.
Point 3: In Introduction, the order of the reference needs to be fixed.
Response 3: All references were fixed and numbered in the order of appearance.
Point 4: In Introduction, (Line 52), please specify which version of ICD you had used. Is it ICD-11 or 10 or what exactly?
Response 4: ICD-10 was used for the description for dental fear as as specific phobia.
Point 5: In Methods, (Line 127), please explain how the questionnaire was conducted. Was it self-administered questionnaire? or is it filled by the investigators? In case it was filled by the investigators, please explain how you calibrated your investigators.
Response 5: Both questionnaires (HAQ and HSPS-G) were self-administered. No investigator filled the questionnaires.
Point 6: In Results, (Line 140), please explain what does Group C refer to?
Response 6: The rapid progession rate (Grade C) according to the 2017 World workshop on the Classification of Periodontal and Peri-implant Diseases and Conditions was explained in the actual manuscipt.
Point 7: In Discussion, please add to your study's limitations the issue of your limited sample size and how it may have led to weak correlation between some variables in your study. for example, the weak correlation between smoking frequency/duration and the periodontitis presence/severity.
Response 7: We added this topic in the discussion. The sample size was given through the number of soldiers with inflammatory periodontal disease sent by 50 military general dentists on different posts to our specialised hospital dental clinic for periodontal treatment for one year of deployment. The number of new patients varies between 80 an 116 per year in the western part of Germany (250 km around Koblenz). The total number soldiers for military deployment during this period of time was about 800-1000 with mandatory dental eximination by general dentists in this year. We reserve treatment time of 4 hours per new patient to get them periodontally dental fit for deployment.
Point 8: In Discussion, there's no need to re-explain the abbreviations of LST and EOE, because they should have been described earlier in the manuscript.
Pesponse 8: We checked all re-explainatation and used the abbreviations of LST and EOE.

Reviewer 2 Report
This is well written paper exploring dental anxiety and sensory processing sensitivity in 116 soldiers during their mandatory oral health examination. The methodology seems sound and statistical methods also appropriate. The results presented indicated higher dental anxiety and sensory processing sensitivity were associated with patients with severe periodontal disease.
The major criticism of this paper as authors already mentioned in this paper is the unique sample subset including only army soldiers. This important issue should be addressed directly in the title of this paper. I suggest to change the title of this paper to reflect the unique subjects used in this research.
Author Response
Resonse to Reviewer 2 Comments
Point 1: The major criticism of this paper as authors already mentioned in this paper is the unique sample subset including only army soldiers. This important issue should be addressed directly in the title of this paper. I suggest to change the title of this paper to reflect the unique subjects used in this research.
Response 1: We changed the word patients with soldiers in the title. So we hope to give this important issue "soldiers" before a unique military deployment a better adress in our research project.

Reviewer 3 Report
The manuscript submitted to IJERPH entitled “Dental anxiety and higher sensory processing sensitivity in patients with inflammatory periodontal disease” is an original article that focus on the evaluation of a possible relationship between dental anxiety and higher levels of SPS in patients (soldiers) with varying degrees of periodontal disease.
On my opinion the article is interesting. Regarding English language, minor spell check is required.
I highlighted some critical issues:
- references are not numbered consecutively [please correct!];
- abbreviations are not clearly defined, therefore I recommend inserting a final paragraph in which they are summarized;
I would also suggest the following change:
In the "Introduction" section on page 1, line 40, I would insert the following sentence:
“Furthermore, periodontitis is a risk factor for the development of osteonecrosis of the jaws in patients taking antiresorptive drugs [DOI: 10.1016/j.jcms.2020.01.014].”
After making these changes, this study may be suitable for publication.
Author Response
Response to Reviewers 3 Comments
Point 1: Regarding English language, minor spell check is required.
Response 1: Final Englisch language spell check was performed 190121
Point 2: references are not numbered consecutively [please correct!];
Response 2: All references were in the submitted manuscipt version 190121 numbered in the order of appearance
Point 3:abbreviations are not clearly defined, therefore I recommend inserting a final paragraph in which they are summarized;
Response 3: At the the end of the manuscipt we added the most important used abbreviations in a final paragraph.
Point 4: In the "Introduction" section on page 1, line 40, I would insert the following sentence:
“Furthermore, periodontitis is a risk factor for the development of osteonecrosis of the jaws in patients taking antiresorptive drugs [DOI: 10.1016/j.jcms.2020.01.014].”
Response 4: Antiresorptive therapy is a exclusion criterion for military deployment to foreign countries. We added the important point in the introduction part as recommended.

Reviewer 4 Report
Congratulations to the Authors, I believe this interesting manuscript merits publication and I have only a few comments and suggestions to them before acceptation. The paper is concise and easy to understand. The statistics are appropriate for this study. The findings are good. The methods, results and discussion sections were all good and clear. The manuscript is well written.
The limits of the study have been highlighted very well and conclusions are appropriate.
About the references, the Authors should check and correct the style and punctuation.
References must be numbered in order of appearance in the text and listed individually at the end of the manuscript.
If other citable research material is available on-line, The Authors may use the website reference style as in guidelines.
The Authors should correct some text inaccuracies:
1) Abstract: line 29 > Define acronym EOE
2) Introduction: line 36 > delete 8,38,41 and follow citation guidelines (square brackets...)
3) Introduction: line 63 > Define acronym PTSD here and not in line 65
4) Discussion: line 253 > representativ > representative
Author Response
Response to Reviewer 4 Comments
Point 1: About the references, the Authors should check and correct the style and punctuation. References must be numbered in order of appearance in the text and listed individually at the end of the manuscript. If other citable research material is available on-line, The Authors may use the website reference style as in guidelines.Introduction: line 36 > delete 8,38,41 and follow citation guidelines (square brackets...)
Response 1: The references were checked, corrected and are now in order of appearance in the text and listed at the end of the manuscript. The recommended reference style is used now.
Point 2: Abstract: line 29 > Define acronym EOE
Response 2: The acronym EOE as ease of excation and the LST (line 27) low sensory threshold were defined in the abstract.
Point 3: Introduction: line 63 > Define acronym PTSD here and not in line 65
Response 3: We defined the acronym PTSD earlier as recommended: Thank you!
Point 4: Discussion: line 253 > representativ > representative
Resonse 4: We changed the german word with the english word "representative". Language and punctuation check was performed 190121.
